# Directions of Immunotherapy for Non-Small-Cell Lung Cancer Treatment: Past, Present and Possible Future

**DOI:** 10.3390/ijms262211055

**Published:** 2025-11-15

**Authors:** Gian Marco Leone, Grazia Scuderi, Paolo Fagone, Katia Mangano

**Affiliations:** Department of Biomedical and Biotechnological Sciences, University of Catania, Via S. Sofia 97, 95123 Catania, Italy; g.marco-94@outlook.it (G.M.L.); graziascuderi@hotmail.it (G.S.); katia.mangano@unict.it (K.M.)

**Keywords:** NSCLC, immunotherapy, immune-checkpoint inhibitors, oncological vaccines, bispecific antibodies

## Abstract

Lung cancer (LC) is one of the most common malignancies and the leading cause of death worldwide. LC is classified into two main histological subtypes: non-small-cell lung cancer (NSCLC), representing the 85% of all LC types, and small-cell lung cancer (SCLC), representing 15% of all lung neoplasm. The recent discovery and clinical approval of new therapeutic approaches has resulted in significant advancements in the management of NSCLC patients. This review aims to summarize the current and ongoing clinical trials that have led to the approval of immune checkpoint inhibitors (ICIs) and the emerging immunotherapy approaches for advanced NSCLC patients. Additionally, the current benefits and drawbacks of these therapeutic strategies will be explored. The treatment for NSCLC is evolving toward a more comprehensive approach that considers both the tumor immune history and genomic features. In this respect, we hope that the ongoing research will make it possible to treat each NSCLC patient individually in the near future.

## 1. Introduction

Lung cancer (LC) is one of the most common malignancies and the leading cause of death worldwide for cancer, in both sexes [1]. Despite the development of new advanced therapies, LC is still a malignancy with an unfavorable prognosis, typically due to the non-specific early symptoms and the lack of a reliable screening test [2,3,4,5].

LC is classified into lung microcitoma or small-cell lung cancer (SCLC) and non-small-cell lung cancer (NSLC), which includes squamous or epidermoid carcinoma (SqCC), adenocarcinoma (ADC), and large cell carcinoma (LCC) [6]. NSCLCs represent around 85% of all LC types, while SCLCs is less common (approximately 15% of lung neoplasms) but characterized by faster growth and a higher propensity for metastasis [7,8,9]. Additionally, rare forms of LC include: adeno-squamous carcinoma, salivary gland-type lung carcinoma, large cell neuroendocrine carcinoma and lung carcinoids [6,10,11].

The gold standard treatment for LC varies based on the type, stage, and molecular profile of the tumor, as well as the patient overall health and preferences [12]. For early-stage NSCLC, surgical resection remains the primary treatment, often followed by adjuvant chemotherapy or radiotherapy to reduce the risk of recurrence [13]. LC outcomes are strongly influenced by the stage at diagnosis. Patients diagnosed at an early stage generally have a favorable prognosis, with 5-year survival rates of approximately 70–90% for Stage I and 50–70% for Stage II [14]. However, survival decreases markedly in more advanced stages, reaching 30–35% for Stage III and only about 9–10% for Stage IV. Unfortunately, LC is most often diagnosed at an advanced stage, when curative options are limited and systemic drug therapy, guided by the immunological and molecular profile of the tumor, represents the standard of care [15,16,17].

Over the last decades, numerous evidence have progressively led to a better understanding of the interactions between cancer cells and the immune system, emphasizing the key role of the immune system in suppressing the development and progression of cancer. The immune response is able to control the growth of tumor cells. In fact, the presence of lymphocytes infiltrating tumor (TILs) is often associated with longer patient survival [18,19]. However, some tumors, known as “cold tumors”, are particularly resistant to the immune response. In particular, these are characterized by the lack of recognizable mutations, an immunosuppressive tumor microenvironment (TME), and the expression of proteins that inhibit the action of T lymphocytes [20].

The advent of immunotherapy has revolutionized the treatment landscape for LC, particularly NSCLC [21,22]. Immune checkpoint inhibitors (ICIs), such as those targeting the PD-1 (programmed death-1) receptor and its ligand PD-L1, have already demonstrated to provide significant improvements in survival for patients with advanced or metastatic NSCLC [12,23]. Drugs like ipilimumab, tremelimumab, pembrolizumab, nivolumab, durvalumab, cemiplimab, and atezolizumab have received both the Food and Drug Administration (FDA) approval for use in various stages of the disease, both as monotherapy and in combination with chemotherapy (Figure 1) [24,25,26,27,28]. These agents work by enhancing the antitumor immune response, releasing the inhibition of T cells exerted by the tumor [29,30]. However, responses are not universal, and primary or acquired resistance and the absence of predictive markers of response remain a challenge [31,32]. Additionally, the optimization of therapeutic combinations and the management of side effects are areas of ongoing research studies. Recent studies are focusing on identifying predictive biomarkers, such as PD-L1 expression, tumor mutational burden (TMB) and Tumor-infiltrating lymphocytes (TILs), and on optimizing therapeutic combinations to further improve the outcomes and to reduce adverse drug reactions [33,34,35]. However, despite their central role in guiding immunotherapy, these biomarkers showed significant limitations. In particular, PD-L1 expression is highly heterogeneous both spatially and temporally, varying between primary and metastatic sites and even within the same tumor region [36]. Its assessment is further complicated by inter-assay variability among immunohistochemical tests (e.g., 22C3, 28-8, SP263, SP142) and the lack of standardized cutoff values, which were historically defined in trial-specific contexts rather than through biologically consistent thresholds [37]. Similarly, TMB, though associated with improved response to immune checkpoint inhibitors, lacks a universally accepted assay or cutoff, and its predictive value may differ depending on sequencing platform, bioinformatic pipeline, and tumor subtype [38,39]. Moreover, TMB does not fully capture the immunogenic landscape, since not all mutations generate neoantigens capable of eliciting an immune response [40]. The use of TILs as a biomarker also faces methodological challenges due to subjective interpretation, intratumoral variability, and absence of validated quantification standards in lung cancer [41]. Collectively, these factors limit the reliability and reproducibility of current biomarkers and underscore the need for integrated, multi-parametric approaches combining molecular, histopathological, and immunological data to better predict immunotherapy outcomes in NSCLC.

This review explores the rationale and current evidence supporting approved immunotherapy strategies for advanced and metastatic NSCLC, integrating insights from preclinical research and ongoing clinical trials. Particular attention is given to emerging combination strategies aimed at overcoming resistance and improving clinical outcomes. Furthermore, novel therapeutic approaches, current challenges, and future perspectives in the evolving landscape of NSCLC immunotherapy will be discussed.

## 2. Immune Checkpoint Inhibitors

Although the immune system and cancer cells coexist, the persistence of these can lead to a depletion of effective anti-tumor T-lymphocytes (“T-cell exhaustion”). In particular, T cells undergo a functional paralysis (“anergy”) leading to the expression of inhibitory immune-checkpoints on their surface, which are engaged by the tumor cells to suppress the immune response [42].

The progress in the understanding of the immunobiology, combined with the recent discoveries on physiological mechanisms that regulate the activity of the immune system and its interactions with the tumor, have allowed development of a new class of therapeutic agents represented by immunomodulatory mAbs capable of blocking ICIs.

Among the various immune checkpoints leveraged by tumors to evade the host immune system, the PD-1/PD-L1 and CTLA-4 pathways are the most well-studied and advanced in clinical applications. Blocking these pathways restores the priming and anti-tumor activity of cytotoxic T cells, which are otherwise inhibited.

### 2.1. PD-1/PD-L1 Axis

The first major breakthrough in the use of ICIs for NSCLC was the FDA approval of nivolumab (a human IgG4 monoclonal antibody (mAb) targeting PD-1) as a second-line therapy for advanced and metastatic NSCLC, in 2015. The approval was based on the results of CheckMate 017 (for SqCC patients) and CheckMate 057 (for non-SqCC patients) phase III trials (Table 1) [26,43]. These trials demonstrated that nivolumab provided superior objective response rates (ORRs) in patients with SqCC and non-SqCC, who had progressed following platinum-based chemotherapy (20% and 19%, respectively) compared to docetaxel group (12.4%). Additionally, patients treated with nivolumab showed a better median OS (9.2 and 12.2 months, respectively) than docetaxel group (6.0 and 9.4 months, respectively), with a 27% to 41% reduction in mortality risk (for non-SqCC and SqCC patients, respectively) [26,43]. Although the studies represented a milestone in shifting NSCLC treatment toward immunotherapy, several limitations should be acknowledged. Biomarker analysis showed that PD-L1 expression was not predictive of clinical benefit in this population, underscoring the complexity of immune response mechanisms and the need for more accurate predictive biomarkers. Additionally, the study design involved a comparison with docetaxel, a drug that is now largely outdated, thereby reducing the long-term relevance of the trial as a reference.

Along the same lines, other ICIs such as pembrolizumab (a humanized IgG4 anti-PD-1 mAb) and atezolizumab (a humanized IgG1 anti-PD-L1 mAb), marked a second breakthrough in the treatment of NSCLC due to their demonstrated efficacy over docetaxel in second-line settings. The success of ICIs in second-line treatment paved the way for their use as first-line therapy for advanced NSCLC.

At the end of 2015, the FDA approved pembrolizumab for the treatment of patients with advanced or metastatic NSCLC expressing PD-L1, following disease progression after platinum-based chemotherapy [44]. Subsequently, in 2016, it was approved as a first-line therapy for patients with advanced NSCLC with PD-L1 expression ≥ 50%, based on the results of the randomized phase III study KEYNOTE-024 (Table 1) [45]. This trial demonstrated that patients treated with pembrolizumab showed a significant improvement in OS rate at 6 months (80.2%) and progression-free survival (PFS) (10.3 months) compared to standard chemotherapy group (72.4% and 6 months, respectively). In addition, Pembrolizumab also demonstrated a higher response rate (44.8% vs. 27.8%) and less frequent adverse drug reaction (73.4% vs. 90.0%) [45]. However, the study population was highly selected, excluding patients with PD-L1 < 50% and those with targetable mutations, which limits generalizability to the broader NSCLC population. Moreover, the reliance on PD-L1 as a single biomarker overlooks the complexity of tumor immunogenicity and the influence of other factors such as tumor mutational burden (TMB) and immune microenvironment. Finally, although long-term follow-up confirmed durable benefit, primary resistance and late relapses remain significant challenges.

Similarly, atezolizumab received the first FDA approval for previously treated patients with metastatic NSCLC, in 2016. The approval was based on the results of the OAK trial (Table 1) [46]. This trial demonstrated that atezolizumab provided a superior OS (15.7 months) compared with chemotherapy alone (10.3 months). In addition, both SqCC and non-SqCC patients with higher PD-L1 expression treated with atezolizumab experienced a longer median OS (20.5 months) [46].

Two years later, a new fully human IgG1 mAb targeting PD-L1 (durvalumab) received the FDA approval to treat patients with unresectable advanced NSCLC, who were previously treated with chemoradiation [47]. Indeed, the results of the PACIFIC study demonstrated that durvalumab increase the median OS (47.5 months versus 29.1 months of placebo) with a 4-year OS rate of 49.6% (for durvalumab arm) compared with 36.3% (for placebo group), and 4-year progression-free survival (PFS) rate of 35.3% compared with 19.5%, respectively (Table 1) [48]. The approval has been extended over time, including combinations with tremelimumab (a human IgG2 mAb targeted CTLA-4) plus chemotherapy for patients with metastatic NSCLC, as demonstrated in POSEIDON, phase III trial [49]. These studies supports durvalumab as a viable first-line option for metastatic NSCLC, particularly in PD-L1–low or negative patients, but its incremental benefit and optimal patient selection require further clarification in future comparative studies. In particular, the magnitude of benefit was modest compared with other contemporary trials, and no clear biomarker-driven subgroup showed a dramatically superior outcome. The addition of tremelimumab increased immune-related adverse events (irAE), raising concerns about the balance between efficacy and toxicity. Additionally, the exclusion of patients with disease progression or poor performance status limits the applicability of results to real-world populations. Moreover, in the evolving landscape of immuno-chemotherapy combinations, the clinical positioning of the durvalumab–tremelimumab regimen remains uncertain, as more established combinations (e.g., nivolumab + ipilimumab) have shown comparable or superior long-term results.

Finally, a human IgG4 mAb targeted PD-1 (cemiplimab) has been approved for the treatment of advanced or metastatic NSCLC with PD-L1 expression ≥ 50%, based on the results of the phase III, EMPOWER-Lung 1 trial, in 2021 (Table 1) [50]. The study compared cemiplimab monotherapy to standard first-line chemotherapy in patients without driver genetic alterations, such as EGFR or ALK. The results demonstrated a significant improvement in OS, with a hazard ratio (HR) of 0.57 (95% CI: 0.42–0.77), indicating a 43% reduction in the risk of death compared to chemotherapy. Additionally, PFS was superior in the cemiplimab group, with an HR of 0.54 (95% CI: 0.43–0.68) [50]. The ORR was 37% in cemiplimab group compared to 21% in patient treated with chemotherapy. One year later, the results from the EMPOWER-Lung 3 phase III study provided further support for the routine administration of cemiplimab plus platinum-based chemotherapy, as a first-line option for NSCLC with high PD-L1 expression (Table 1) [51]. Although the EMPOWER-Lung 3 trial demonstrated significant improvements in OS and PFS with cemiplimab plus chemotherapy, several limitations should be considered. The study included patients with any level of PD-L1 expression, but it was not powered to provide definitive conclusions for subgroups such as PD-L1–negative patients, leaving uncertainty about the magnitude of benefit in these populations [52]. Additionally, early trial termination for efficacy, while ethically justified, may introduce bias and affect the maturity of the long-term data. Although quality-of-life outcomes favored the combination, the magnitude of improvement was modest, and the trial was not primarily designed to assess these endpoints. Finally, longer follow-up is needed to fully characterize the durability of responses and the long-term safety profile, particularly regarding irAE that may emerge with extended treatment.

However, despite the clinical success of ICIs reported above, most NSCLC patients, treated with a single anti PD-1/PD-L1 agent still fail to initially respond to treatment (primary resistance) or experience disease progression after an initial response (acquired resistance). The potential mechanisms underlying these two types of resistances include the lack of immune cell infiltration in the TME, aberrant PD-L1 expression, and genetic or epigenetic alterations that affect tumor antigen presentation [53,54,55]. Additionally, some tumor cells may exploit alternative immune inhibition pathways or create an immunosuppressive microenvironment [56]. To address these challenges, research is focusing on innovative strategies, such as combining ICIs with targeted therapies, anti-angiogenic agents, chemotherapy, or novel ICIs or immunotherapies strategies, to overcome resistance mechanisms and enhance treatment efficacy [53,56].

**Table 1 ijms-26-11055-t001:** Overview of Clinical Trial Designs.

Study	Enrollment	Endpoints	Phase	Ref.
AMBER	219	Safety and tolerability Maximum tolerated dose (MTD) Recommended phase II dose (RP2D)	Phase 1	[57]
CA184-156	954	OS, PFS, ORR, irPFS and irAEs	Phase 3	[58]
CHECKMATE-9LA	719	OS, PFS, ORR and DOR	Phase 3	[59,60]
CheckMate-017	272	OS, PFS, ORR and DOR	Phase 3	[26]
CHECKMATE-227	1739	OS, PFS, ORR and DOR	Phase 3	[61]
CITYSCAPE	135	OS, PFS, ORR, DOR and AE	Phase 2/3	[62]
EMPOWER-Lung 1	712	OS and PFS	Phase 3	[51]
EMPOWER-Lung 3	466	OS, PFS and ORR	Phase 3	[52]
HARMONi-2	398	PFS	Phase 3	[63]
Impower132	578	OS, PFS, ORR and DOR	Phase 3	[64]
KEYNOTE-024	305	OS and PFS	Phase 3	[45]
KEYNOTE-189	616	OS and PFS	Phase 3	[65]
KEYNOTE-407	559	OS and PFS	Phase 3	[66]
NEOpredict-Lung	60	Surgery within 43 days, OS and DFS	Phase 2	[67]
OAK	1225	OS	Phase 3	[46]
RELATIVITY-047	714	OS, PFS and ORR	Phase 3	[68]
SKYSCRAPER-01	534	OS and PFS	Phase 3	[69]

### 2.2. CTLA-4 Pathway

Cytotoxic T-Lymphocyte Antigen-4 (CTLA-4) is a key immune checkpoint that regulates the amplitude and duration of T-cell activation. It competes with the co-stimulatory receptor CD28 for binding to B7 ligands (CD80/CD86) on antigen-presenting cells (APCs), thereby limiting T-cell proliferation and preventing excessive immune activation [70]. Ipilimumab, a monoclonal antibody targeting CTLA-4, was the first immune checkpoint inhibitor (ICI) to receive FDA approval, demonstrating significant clinical benefit in patients with advanced melanoma compared to the gp100 peptide vaccine [70]. Another anti-CTLA-4 antibody, tremelimumab, remains under clinical evaluation for several solid tumors, including LC [71]. By blocking the CTLA-4/B7 interaction, these antibodies enhance T-cell activation and cytokine production, thereby promoting cytotoxic activity within the TME [72].

Following the success of ipilimumab in melanoma, several trials explored its potential role in NSCLC and SCLC. A Phase II randomized, double-blind study evaluated ipilimumab in combination with chemotherapy as a first-line treatment for both NSCLC and SCLC [73]. The study demonstrated modest improvements in immune-related progression-free survival (irPFS) and OS compared to chemotherapy alone (median OS 12.2 vs. 8.3 months). However, these improvements did not consistently translate into statistically significant clinical benefit, and toxicity remained a concern. In particular, irAEs of grade 3–4 occurred in 15–20% of patients receiving ipilimumab.

In contrast, the Phase III CA184-156 trial yielded disappointing results (Table 1). Among 476 patients with advanced SCLC treated with ipilimumab plus chemotherapy, no significant improvement was observed in either OS (11.0 vs. 10.9 months) or PFS (4.6 vs. 4.4 months) compared with placebo [58]. Moreover, 27% of patients experienced treatment-related immune toxicities, underscoring the narrow therapeutic window of CTLA-4 blockade in this setting.

Collectively, these studies highlight a translational gap between the early promise of CTLA-4 inhibition observed in melanoma and its limited success in lung cancer. Although ipilimumab and tremelimumab remain under investigation in combination with PD-1/PD-L1 inhibitors and chemotherapy, the risk–benefit ratio appears less favourable in unselected NSCLC and SCLC populations. The lack of significant survival benefit in pivotal Phase III trials suggests that CTLA-4 inhibition alone may not be sufficient to overcome the profound immunosuppressive milieu of lung tumors.

Future research should focus on biomarker-driven patient selection, optimal dosing and sequencing strategies, and combination regimens that balance efficacy with manageable toxicity. A deeper understanding of TME heterogeneity, T-cell exhaustion, and irAE mechanisms will be essential to refine the therapeutic positioning of CTLA-4 blockade within the evolving landscape of lung cancer immunotherapy.

## 3. Combination Therapy

Monotherapy with ICIs has significantly improved survival outcomes for patients with NSCLC, revolutionizing the treatment of these patients. However, several patients do not benefit significantly from immunotherapy alone [74]. For these patients, it has been hypothesized that chemotherapy, traditionally used to reduce tumor burden by inducing DNA damage and apoptosis in tumor cells, may play a key role in modifying the TME [75]. In this context, chemotherapy acts not only as a cytotoxic agent but also as a catalyst that renders the tumor immunogenic, thereby optimizing the efficacy of ICIs.

The understanding of the synergistic potential between chemotherapy and ICIs provided the theoretical foundation for the design of several studies including the two trials KEYNOTE-189 and KEYNOTE-407, which revolutionized first-line treatment for patients with metastatic NSCLC (Table 1). These studies evaluated the efficacy of combining pembrolizumab with standard chemotherapy regimens, demonstrating significant improvements in OS and PFS compared to chemotherapy alone, regardless of PD-L1 expression levels [65,66].

In particular, the KEYNOTE-189 trial included patients with non-SqCC, using a combination of pembrolizumab with pemetrexed and platinum-based chemotherapy, while KEYNOTE-407 focused on patients with SqCC, combining pembrolizumab with carboplatin and paclitaxel (or nab-paclitaxel). The results of both studies demonstrated that the chemotherapy–pembrolizumab combination not only improved the duration of response (DOR) but also significantly reduced the risk of death by 44%, and doubled PFS, extending it from 4.9 months to 9 months [65,66]. Importantly, the observed benefits were independent of PD-L1 expression levels, demonstrating efficacy across all subgroups, including those with PD-L1 < 1%, PD-L1 between 1–49%, and PD-L1 ≥ 50%. These landmark results led to the FDA approval of the chemotherapy–pembrolizumab combination for first-line treatment, prompting updates to international guidelines and establishing this approach as the new standard of care for patients with metastatic NSCLC, in 2018 [76]. Despite the trial clear clinical impact, several limitations should be acknowledged. The study excluded patients with targetable driver mutations and those with poor performance status, limiting the generalizability of the results to these populations. While PD-L1 expression was assessed, the trial was not powered to determine definitive benefits in PD-L1–low or negative subgroups. Furthermore, the control arm consisted of chemotherapy alone, which, given the current availability of multiple immunotherapy combinations, reduces its contemporary relevance. Finally, although long-term follow-up demonstrated durable survival benefits, extended monitoring is needed to fully characterize late irAE and long-term safety.

On the other hand, the results of the Impower132 clinical study demonstrated that patients treated with atezolizumab in combination with carboplatin/cisplatin and paclitaxel showed significant benefits in PFS with a median PFS of 7.6 months compared to 5.2 months for chemotherapy alone (Table 1) [64]. However, the trial did not demonstrate a statistically significant improvement in OS, with median OS reaching 18.1 months in the combined arm versus 13.6 months in the chemotherapy group. Despite the lack of OS benefit, the combination was well tolerated, with manageable safety profiles consistent with those of the individual agents [64,77]. However, the health-related quality-of-life data were limited, making it difficult to assess the broader patient-centered impact of the treatment. Finally, the study did not explore optimal sequencing strategies for patients who progress after the combination, leaving unanswered questions about subsequent therapy choices.

Several randomized trials have demonstrated that combining anti-CTLA-4 with anti-PD-1 mAbs offers greater clinical benefits compared to chemotherapy or single-agent therapies. In particular, among these two randomized clinical trials (CHECKMATE-227 and CHECKMATE-9LA) showed better efficacy of ipilimumab (a fully human IgG1 mAb targeting CTLA-4) plus nivolumab treatment compared with chemotherapy alone, allowing the FDA approval of the dual checkpoint blockade to treat patients with advanced NSCLC patients, in 2020 (Table 1) [78].

In particular, the CHECKMATE-9LA investigated the efficacy of ipilimumab plus nivolumab and two cycles of chemotherapy versus chemotherapy alone [59]. NSCLC patients treated with the combination therapy showed a median OS rate of 14.1 months compared with 10.7 months of patients treated with chemotherapy alone, with an overall survival rate of 38% versus 25%, respectively [59]. Further analyses showed that after a minimum follow-up of 68.6 months, the combination regimen demonstrated a sustained OS benefit compared with chemotherapy alone (HR = 0.74; 95% CI 0.63–0.87), with 6-year OS rates of 16% vs. 10%, respectively. The survival advantage was consistent across PD-L1 expression levels (<1%: 20% vs. 7%; ≥1%: 15% vs. 10%) and histologic subtypes (SqCC: 14% vs. 5%; non-SqCC: 17% vs. 12%) [60]. The 6-year PFS rates were 9% in the nivolumab plus ipilimumab arm versus 3% with chemotherapy alone (HR = 0.70; 95% CI 0.59–0.82). The DOR was notably longer with the immunotherapy-based regimen, with 19% of patients maintaining a response at 6 years, whereas all responses in the chemotherapy arm had ceased by that time. Exploratory genomic analyses indicated that patients harboring KRAS, STK11, KEAP1, or TP53 mutations also derived clinical benefit, confirming the durable efficacy and manageable safety profile of nivolumab plus ipilimumab combined with limited chemotherapy as a first-line regimen for metastatic NSCLC [60].

On the other hand, the CHECKMATE-227 trial evaluated the safety profile of nivolumab plus ipilimumab as first-line treatment compared to standard chemotherapy. The study was conducted in two parts, focusing on patients with high TMB and those stratified by PD-L1 expression [61]. In the cohort with high TMB, the combination treatment demonstrated a significant improvement in OS compared to chemotherapy, with a median OS of 17.1 months versus 14.9 months, respectively [61]. Moreover, long-term follow-up showed durable survival benefits, with a 5-year OS rate of 24% in the combination arm compared to 14% with chemotherapy [79]. Overall, the trial provided landmark evidence supporting dual checkpoint inhibition in advanced NSCLC and expanded understanding of immunotherapy beyond PD-L1 high patients, but its complex design, heterogeneous outcomes, and safety considerations highlight the need for careful patient selection and further studies to optimize therapeutic strategies.

A recent pooled analysis of the two previously described phase 3 studies, highlighted the long-term benefits of first-line treatment with nivolumab plus ipilimumab, with or without two cycles of chemotherapy, compared to chemotherapy alone in patients with metastatic NSCLC and low PD-L1 expression (<1%). The results confirm that this immunotherapy combination offers a durable and clinically meaningful benefit in a population with still unmet therapeutic needs. Indeed, among the 322 patients treated with the combination immunotherapy, the median OS was 17.4 months, significantly longer than the 11.3 months observed in the chemotherapy arm (HR = 0.64), with a 5-year OS rate of 20% versus 7%. The benefit was consistent across particularly challenging subgroups, such as patients with brain metastases (HR = 0.44) or squamous NSCLC (HR = 0.51). In the overall population, the median PFS was 5.4 months with nivolumab/ipilimumab, compared to 4.9 months with chemotherapy, and the median DOR was 18.0 months versus 4.6 months [80].

Efforts are also underway to combine immune checkpoint inhibitors (ICIs) with targeted therapies, including FDA-approved anti-VEGF monoclonal antibodies (mAbs) such as ranibizumab and bevacizumab. These agents are being explored for their potential to enhance immunomodulation while exerting both anti-angiogenic and immune-modulating effects on tumor cells. This combinatorial approach aims to investigate potential additive or synergistic effects, improving therapeutic efficacy against tumors [81,82,83].

The use of bevacizumab in the treatment of lung neoplasms could represent a valid therapeutic opportunity to overcome tumor resistance to monotherapy [84]. Indeed, anti-VEGF therapies help stabilize newly formed blood vessels within the tumor, reducing abnormal vascular permeability. This stabilization enhances the infiltration of lymphocytes into the TME, potentially improving the efficacy of immune responses and immunotherapy [85].

Although the ICIs represent an effective treatment on a broad spectrum of NSCLC, their utility in the therapeutic algorithm for “oncogenic-addicted” tumors is yet to be demonstrated. However, it has been reported that the overexpression of PD-L1 with TPS (Tumor Proportion Score) > 50% of tumor cells can be observed in tumors with ROS1 rearrangement and, less frequently, in tumors with rearrangements of ALK [86,87,88]. Occasionally, EGFR-mutated tumors may also exhibit marked expression of PD-L1 [89]. In these cases, the overexpression of PD-L1 does not allow the eligibility of the patient for ICI, since the latter may be ineffective and inhibitors of great benefit tyrosine kinases. Overall, it is clear how the assessment of PD-L1 expression should be included as part of a more elaborate diagnostic/therapeutic algorithm.

## 4. Novel Immune Checkpoint Targets

Although ICIs have revolutionized the treatment of NSCLC, there remains significant potential to enhance their efficacy and broaden their utility. Despite their success in a subset of patients, many individuals remain ineligible, fail to respond, or eventually develop resistance to therapies targeting CTLA-4 and PD-1/PD-L1 pathways [90].

Resistance to CTLA-4 or PD-1/PD-L1 blockades remains a major barrier in the treatment of NSCLC. A significant proportion of patients exhibit primary resistance, meaning they derive minimal or no benefit from the outset, while others develop acquired resistance after an initial response. Mechanistically, primary resistance often stems from tumor–immune escape at multiple levels of the cancer–immunity cycle: deficient antigen presentation (e.g., downregulation of HLA-I or loss of β2-microglobulin), oncogenic pathway activation (e.g., STK11, KEAP1, EGFR/ALK aberrations) and an immunosuppressive microenvironment with low T-cell infiltration or high myeloid-derived suppressor cell (MDSC) load [91].

On the other hand, acquired resistance may result from clonal selection of tumor sub-populations lacking neoantigens, epigenetic or signaling pathway changes (e.g., JAK1/JAK2 mutations, IFN-γ pathway defects), upregulation of alternative immune checkpoints (e.g., TIM-3, LAG-3) and remodelling of the tumor microenvironment to a “cold” phenotype. Translational investigations in NSCLC have, for example, documented acquired loss of HLA class I expression with matched pre- and post-treatment biopsies, and reduced CD8^+^ T-cell infiltration at progression [92]. These mechanistic insights are now informing therapeutic strategies to overcome resistance: explore and identify alternative immune checkpoints, such as TIGIT, LAG-3, and TIM-3, dual checkpoint blockade (CTLA-4 + PD-1 or PD-1 + LAG-3/TIGIT), modulation of antigen presentation (e.g., epigenetic agents restoring HLA-I), and targeted therapy of co-mutated genes (e.g., STK11/KEAP1) [93,94,95,96,97].

Despite this progress, many of these strategies remain in early-phase trials and lack definitive clinical validation. Future research must prioritise predictive biomarkers of resistance, longitudinal monitoring of immune phenotypes, and rational combinations tailored to the molecular and immunologic profile of each patient’s tumor.

### 4.1. Lymphocyte Activation Gene 3

LAG-3 (Lymphocyte-activation gene 3) is a surface protein expressed on effector T cells and regulatory T cells (Tregs), playing a key role in modulating the adaptive immune response. It is encoded by the LAG3 gene, which is localized on the short arm of chromosome 12 [98]. The interaction between LAG3 and its ligands (Major Histocompatibility Complex II (MHC-II) or Fibrinogen-like protein 1 (FGL1)) leads to reduce T cell function and cytokines production (inhibiting T cell receptor (TCR) signaling), contributing to the maintenance of immune homeostasis (Figure 2) [99]. Recently, LAG-3 has been identified as an important immune checkpoint in different cancer types, and its blockade has been associated with enhanced antitumor responses.

Although anti-LAG3 monotherapy has generally demonstrated limited antitumor efficacy, the combination of anti-LAG3 and anti-PD-1 has demonstrated synergistic activity in tumor models, leading to the development of the first anti-LAG3 mAbs, relatlimab [100].

Relatlimab is a humanized IgG4 mAb targeting LAG-3. In the RELATIVITY-047 trial, the combination of relatlimab plus nivolumab showed for the first time a significant improvement in PFS compared to nivolumab monotherapy in metastatic melanoma (10.1 months compared with 4.6 months, respectively) [101]. These findings led to the FDA approval of relatlimab in March 2022, marking it as the first anti-LAG3 therapy approved for use in metastatic melanoma [68].

Based on this rationale, relatlimab combined with nivolumab as neoadjuvant versus nivolumab monotherapy is under evaluation to treat resectable NSCLC patients, in the NEOpredict-Lung trial. The study showed a partial response rate of 27% in the combination arm versus 10% in nivolumab alone in patients with a positive expression of PD-L1 [67].

Alternative approaches, such as eftilagimod alpha, a soluble LAG-3 protein activating antigen-presenting cells (APCs), have shown promise in combination with pembrolizumab, particularly in PD-L1-high NSCLC patients, with 8.3% ORR and 33.3% Disease Control Rate (DCR) [102]. As LAG-3 inhibitors, such as relatlimab and eftilagimod, continue to reshape therapeutic paradigms, their integration with other immune-modulating agents holds great potential to advance the treatment of cancers.

Recently, several ongoing studies have identified other LAG-3 inhibitors (favezelimab, fianlimab, GLS-012, HLX26, and LBL-007), which have been tested as monotherapy or in combination with other ICIs in advanced solid tumors (Table 2). Although some of these agents exhibited good tolerability, they failed to demonstrate significant efficacy, especially in heavily pretreated or ICI-resistant populations [103]. These results highlight the challenges in effectively targeting the LAG-3 pathway within the complex immunosuppressive environment of NSCLC. Consequently, there is a pressing need to develop innovative strategies to inhibit LAG-3, including optimizing combination regimens with other ICIs, exploring novel biomarkers to select responsive patients, and designing next-generation LAG-3-targeted agents to overcome resistance mechanisms.

### 4.2. T Cell Immunoglobulin and Mucin-Domain-Containing-3

The T cell immunoglobulin and mucin-domain-containing-3 (TIM-3) gene is located on the long arm of chromosome 5. It encodes for a membrane protein that acts as an immune checkpoint receptor [104]. The protein is expressed in various immune cells, including T cells, natural killer (NK) cells, dendritic cells, and monocytes [105]. TIM-3 plays a crucial role in maintaining immune tolerance and regulating the immune response by inhibiting the activation and proliferation of T cells [106]. It interacts with multiple ligands, such as galectin-9, phosphatidylserine, HMGB1, and CEACAM1, to modulate immune activity in physiological and pathological contexts (Figure 2). In cancer, TIM-3 expression is often upregulated in exhausted T cells, contributing to immune evasion by tumors [105]. As a result, TIM-3 has emerged as a promising therapeutic target, with ongoing research focusing on its blockade to restore T cell function and enhance antitumor immunity. Recent phase 1 clinical trials have highlighted the potential of TIM-3 blockade in the treatment of NSCLC, particularly in combination with PD-1/PD-L1 immune checkpoint inhibitors, to improve outcomes in different cancer types.

In this context, cobolimab, a mAb of the IgG4 subclass targeting TIM-3, demonstrated a favorable safety profile both as monotherapy and in combination with dostarlimab (an anti-PD-1 agent), although its initial antitumor activity in NSCLC patients was limited [107]. In particular, the phase 1 AMBER study evaluated the combination of cobolimab and dostarlimab in patients with advanced or metastatic NSCLC previously treated with anti-PD(L)-1 therapy (Table 1). The aim was to explore a potential strategy to overcome acquired resistance to immunotherapy. The results showed an ORR of 8.3%, while the disease control rate DCR was 21.4%, indicating limited but potentially meaningful clinical activity in a heavily pretreated population. The safety profile of the combination was manageable, with adverse events mostly mild to moderate in severity. These preliminary data suggest that adding TIM-3 blockade to PD-1 inhibition could represent a promising approach to enhance responses in NSCLC patients refractory to standard immunotherapy, supporting further investigation in selected cohorts or in combination with other agents [57].

Recently, another humanized IgG4 mAb designed to block TIM-3 developed by Novartis, sabatolimab, was tested in combination with spartalizumab (a humanized IgG4 mAb targeting PD-1) in patients with NSCLC who had previously received anti-PD-1/L1 therapies. The combined approach demonstrated limited efficacy, with only 35% of patients achieving a stable response, while the majority experienced disease progression [108]. Despite its tolerability in combination, the modest results highlighted the need for further optimization of TIM-3 blockade in NSCLC. These findings support further investigation into TIM-3 as a therapeutic target to overcome resistance to PD-1/PD-L1 inhibitors. Current research focuses on improving the therapeutic impact of sabatolimab and other new anti-TIM-3 mAbs, through improved combination strategies or biomarker-based patient selection, to better address the challenges of immune resistance in NSCLC (Table 3).

### 4.3. T Cell Immunoreceptor with Immunoglobulin and ITIM Domain

T cell immunoreceptor with immunoglobulin and ITIM domain (TIGIT) gene is located on the long arm of chromosome 3. It encodes for a protein that acts as an immune checkpoint receptor expressed on T cells and natural killer (NK) cells, playing a critical role in suppressing anti-tumor immune responses [109]. The binding between TIGIT and its ligands CD155 (PVR) and CD112 is involved in the reduction of CD226 and TCR signaling pathway, which are essential for the activation of effector T cells (Figure 2) [110]. In particular, TIGIT competes with CD226 for binding to ligands expressed on tumor cells and antigen-presenting cells (APCs). This competition is crucial for regulating the immune response, as CD226 binding to ligands promotes homodimer formation and the activation of T and NK cells. In contrast, TIGIT prevents this interaction and recruits inhibitory signals through the ITIM domain, reducing immune cell proliferation and effector function [110]. Moreover, the interaction of TIGIT with CD155 on APCs can induce polarization toward a tolerogenic phenotype, further contributing to immunosuppression in the TME [111].

In NSCLC, overexpression of PVR and increased TIGIT expression correlate with impaired T cell activation, promoting immune exhaustion and reduced cytotoxicity [112]. This imbalance suppresses antitumor immunity and facilitates immune evasion. Targeting this axis, particularly through TIGIT blockade, is a promising therapeutic strategy aimed at restoring T cell function and enhancing the efficacy of immunotherapy in NSCLC [112].

Recently, it was suggested that combining TIGIT blockade with PD-1/PD-L1 inhibitors could facilitate the differentiation of activated T cells into effector or memory T cells, thereby boosting anti-tumor immune responses [113]. Clinical benefit has been associated with CD226 expression in patients with NSCLC treated with the anti-PD-L1 mAbs atezolizumab [22]. Mechanistically, PD-1 suppresses the phosphorylation of CD226 and CD28 through its ITIM-containing intracellular domain (ICD) [114]. Concurrently, TIGIT limits CD226 co-stimulation by blocking its interaction with the shared ligands. To fully restore CD226 signaling and maximize immune anti-tumor activity, simultaneous inhibition of both TIGIT and PD-1 pathways is required, providing a solid mechanistic basis for dual targeting in clinical applications [115]. The preliminary efficacy of the combination therapy involving the novel anti-TIGIT mAb, tiragolumab, and atezolizumab was evaluated for chemotherapy-naive patients with PD-L1 positive metastatic NSCLC, in the CITYSCAPE trial (Table 1) [62]. The trial aimed to assess the potential synergistic effect of blocking both TIGIT and PD-L1 pathways to enhance immune responses against tumors. In 2022, the preliminary results of this trial showed promising effects, with the combination therapy leading to an improved ORR and PFS compared to atezolizumab alone (31.3% vs. 16.2% and 5.4 vs. 3.6 months, respectively) [62]. Specifically, the combination demonstrated a higher clinical benefit in patients with high PD-L1 expression, suggesting that targeting multiple immune checkpoints may overcome resistance mechanisms observed with single-agent immunotherapy. These findings highlight the potential of dual checkpoint inhibition as a strategy to improve outcomes in NSCLC, particularly in patients who may not respond optimally to current therapies.

Despite the initial promise of TIGIT blockade, recent Phase II/III results have been less encouraging. Notably, The Phase III SKYSCRAPER-01 trial evaluated the combination of the anti-TIGIT monoclonal antibody tiragolumab with the anti–PD-L1 mAb atezolizumab as first-line therapy for patients with PD-L1–high (≥50%) advanced or metastatic NSCLC [69]. Although the combination showed a numerical improvement in both PFS (median 7.0 vs. 5.6 months; HR = 0.78) and OS (median 23.1 vs. 16.9 months; HR = 0.87) compared with atezolizumab monotherapy, these differences did not reach statistical significance [69]. Similarly, subsequent analyses did not confirm the early efficacy signals observed in CITYSCAPE study. Indeed, in 2024 the Phase II/III SKYSCRAPER-06 trial failed to demonstrate a statistically significant improvement in OS or PFS with tiragolumab plus atezolizumab and chemotherapy compared to pembrolizumab in combination with chemotherapy in non-SqCC NSCLC patients [116]. These outcomes have tempered the initial enthusiasm for TIGIT-targeted immunotherapy, underscoring the complexity of checkpoint interactions and the need for refined biomarker-driven strategies to identify responders.

Recently, other newly developed anti-TIGIT mAbs including four humanized IgG1 mAb targeted TIGIT (domvanalimab, ociperlimab, rilvegotomig, and tiragolumab) are under evaluation in several phase I/II studies for NSCLC patients (Table 4). Early findings indicate that these mAbs in combination with PD-1 inhibitors achieve higher response rates compared to PD-1/PD-L1 inhibition alone, likely driven by synergistic mechanisms such as enhanced activation of NK cells and CD8+ tumor-infiltrating lymphocytes (TILs). However, further investigations are necessary to establish the optimal sequencing of these therapies and to identify patient subgroups that would benefit most from early integration of chemotherapy.

In summary, while the TIGIT pathway remains a biologically compelling target for immune modulation in NSCLC, its clinical validation is still evolving. Future research should focus on unravelling mechanisms of primary and adaptive resistance, optimizing combination regimens, and stratifying patients based on CD226 expression, PD-L1 levels, and tumor immune contexture to fully harness the therapeutic potential of TIGIT inhibition.

### 4.4. Novel Potential Immune Checkpoints

Recently, the exploration of novel immune checkpoints, such as IDO1, GITR, NKG2A, CD38, CD47, OX-40 and adenosine receptors, is gaining attention as potential targets to overcome resistance to conventional ICIs. Recent preclinical and clinical studies are actively evaluating novel agents directed against these checkpoints to enhance anti-tumor immunity [117]. For example, targeting CD47, a “don’t eat me” signal that prevents macrophage-mediated phagocytosis, has shown promise in preclinical models and early-phase trials [118]. In a Phase I clinical trial evaluating the IgG4 anti-CD47 antibody magrolimab in combination with chemotherapy, encouraging preliminary responses were observed in advanced NSCLC patients, suggesting enhanced tumor clearance [119]. Similarly, inhibition of NKG2A, an inhibitory receptor expressed on NK cells and cytotoxic T cells, with the IgG4 mAb monalizumab in combination with durvalumab demonstrated improved response rates in early-stage studies, particularly in PD-L1-positive NSCLC [120]. Additionally, targeting the adenosine pathway, which suppresses immune responses in the TME, with adenosine receptor antagonists (A2AR) have shown promise in combination with PD-1 inhibitors in early-phase clinical trials [121,122]. Although most of these novel strategies are still in their early development stages, preliminary data highlight their potential to provide effective therapeutic options and further broaden the scope of immunotherapy for NSCLC treatment. Larger clinical studies will be essential to confirm their efficacy and safety profiles, paving the way for the integration of these approaches into future treatment regimens.

## 5. Novel Approaches to NSCLC Immunotherapy

The use of ICIs has revolutionized the treatment of many cancer types, including NSCLC. However, their use is associated with irAEs, which result from immune system activation against healthy tissues [123]. IrAEs can affect various organs and range in severity from mild to potentially life-threatening [124]. Their severity ranges from mild (grade 1–2) to potentially life-threatening (grade 3–4), with reported incidence in NSCLC patients of up to 60–70% for any-grade irAEs and approximately 10–15% for severe irAEs requiring hospitalization or immunosuppressive therapy [125,126].

Early recognition and proactive management are essential to minimize complications while preserving the anti-tumor efficacy of ICIs. Guidelines recommend regular monitoring, patient education, and prompt intervention with corticosteroids or other immunosuppressive agents, such as infliximab, for high-grade events [127]. Importantly, several studies indicate that appropriate management of irAEs does not necessarily compromise therapeutic benefit [128]. In fact, in some cases, the occurrence of certain irAEs, such as dermatologic or endocrine events, has been correlated with improved clinical outcomes, suggesting a complex relationship between immune activation and tumor control [129]. However, some data indicate that adherence to these recommendations remains suboptimal, with less than half of irAEs managed according to established protocols [130]. This highlights the need for improved clinician education and standardized documentation. Innovative strategies—such as nurse-led active monitoring and multidisciplinary supportive care—are emerging as valuable tools to enable earlier intervention, enhance patient safety, and promote a more patient-centered approach to immunotherapy management.

Emerging next-generation immunotherapy approaches in NSCLC aim to expand the therapeutic potential of immune-based treatments beyond traditional ICIs. Novel strategies include therapeutic agents like bispecific mAbs, personalized cancer vaccines and adoptive T-cell therapies, including the chimeric antigen receptor (CAR-T) cells engineered to recognize tumor-specific antigens, are advancing with promising preclinical and early clinical results in NSCLC [131,132]. These emerging therapies, combined with existing ICIs or used in novel combinations, hold great promise for improving outcomes in patients with advanced NSCLC, particularly those who are refractory to current immunotherapies [133].

### 5.1. Adoptive T Cell Tranfer

Adoptive T-cell therapy (ACT) is an advanced and innovative approach in cancer immunotherapy, using the patient’s immune cells to more effectively fight malignancies. This process involves extracting T lymphocytes from the patient, enhancing T cells antitumor properties through laboratory modification or expansion, and reinfusing them in the same patient to specific target and eliminate cancer cells [134]. The ACT encompasses different techniques, such as employing tumor-infiltrating lymphocytes (TILs), engineering T cells to express chimeric antigen receptors (CAR-T cells), or modifying them with tumor-specific T-cell receptors (TCRs).

Tumor-infiltrating lymphocytes (TILs), one of the most classic forms of ACT, exploiting the intrinsic ability of T cells to recognize and attack tumor antigens present within the TME. Although this strategy has shown promising results in specific cancer types, such as melanoma, the complexity of NSCLC poses significant challenges [135,136]. Indeed, the pronounced antigenic heterogeneity of NSCLC presents a substantial obstacle to identifying a uniform therapeutic target. Furthermore, the highly immunosuppressive TME significantly impairs the effectiveness of T-cell responses [137]. To overcome these limitations, research has led to the development of more advanced technologies, such as genetically engineered T cells, including CAR-T cells and TCR-engineered T cells, offering new perspective in the treatment of NSCLC. In particular, advancements in gene recombination techniques have led to the development of “super-lymphocytes” genetically engineered to enhance their specificity toward antigens overexpressed in tumors, even in the absence of natural reinforcement by APCs [138]. Specifically, T lymphocytes from the patient are extracted and undergo genetic engineering to express artificial antigen receptors known as “Chimeric Antigen Receptors” (CARs) (Figure 3). The receptor is composed of a single-chain variable fragment (scFv) derived from an antibody, which recognizes a specific tumor-associated antigen, fused with intracellular signaling domains that activate T-cell responses [139]. CARs are designed to bypass the requirement for antigen presentation by MHC molecules, allowing T cells to target tumor cells directly. Additionally, the receptor typically includes a costimulatory domain, such as CD28 or 4-1BB, to enhance the persistence and efficacy of the T cells [140].

The highly innovative CAR-T technology was initially developed by the University of Pennsylvania [141,142]. It has demonstrated remarkable success in hematologic malignancies and is being actively investigated for its potential for solid tumors including NSCLC [143]. In this regard, the application of CAR-T cell therapy in NSCLC has shown promising preliminary results in terms of safety, but its clinical efficacy remains limited. Several phase I and I/II trials have confirmed the feasibility and tolerability of this approach, reporting a low incidence of severe adverse events and only rare cases of cytokine release syndrome or pulmonary toxicity, which were generally manageable with corticosteroids. In the trial NCT01869166 on EGFR-CAR-T cells, 11 patients with metastatic EGFR-positive NSCLC received a single CAR-T infusion after lymphodepletion: no treatment-related serious AEs were observed, but only two partial responses and five cases of stable disease were reported, with a median PFS of 7 weeks, indicating transient antitumor activity [144].

From an engineering standpoint, innovations have sought to overcome the biological barriers typical of solid tumors. The use of non-viral systems (such as piggyBac) has streamlined manufacturing. In particular, the phase I clinical trial NCT03182816 evaluated the safety and feasibility of piggyBac-generated EGFR-targeted CAR-T cells in patients with advanced, relapsed/refractory EGFR-positive NSCLC. The therapy was well tolerated, with only mild to moderate fever (grade 1–3) as the most common AE and no severe cytokine release syndrome (CRS) reported [145]. CAR-T cells were detectable in the peripheral blood of eight patients after infusion; one patient achieved a durable partial response lasting over 13 months, six had stable disease, and two experienced disease progression. The median PFS was 7.1 months, and the median OS was 15.6 months, suggesting a good tolerability and promising clinical activity [145]. These findings support further investigation of piggyBac-engineered CAR-T therapy as a simpler and more cost-effective alternative to viral vector-based CAR-T cell generation in NSCLC.

Emerging targets like CD276 and PD-L1 have also gained attention for their diagnostic and immunological relevance in NSCLC [146]. CAR-T cells designed to target PD-L1 have exhibited potent efficacy against tumors with both high and low PD-L1 expression. Innovative strategies, such as combining CAR-T therapy with localized irradiation or modifying CAR-T cells to express receptors like CXCR5 for enhanced tumor infiltration, are being explored to overcome the challenges posed by the NSCLC immunosuppressive TME and antigenic heterogeneity [147]. Despite these advances, CAR-T therapy for NSCLC is still in its early stages, requiring further research to address hurdles like tumor antigen escape, limited tumor infiltration, and AEs, before it can achieve widespread clinical application.

Recently, new generations of “armored CAR-T cells,” designed to secrete immunostimulatory cytokines (IL-12, IL-7, IL-15, CCL19) or incorporating second-generation costimulatory domains (CD28, 4-1BB), have shown improved tumor infiltration and persistence in preclinical models [148]. However, significant challenges remain, including poor trafficking into the tumor microenvironment, antigenic heterogeneity leading to immune escape, limited in vivo persistence, and potential on-target/off-tumor toxicity due to antigen expression in normal tissues.

Despite a high rate of clinical responses, CAR-T therapy is unfortunately accompanied by several limits. One major challenge is the high cost and complexity of manufacturing CAR-T cells, which requires personalized cell engineering for each patient. Another critical limitation is the potential for severe side effects, including CRS and immune effector cell-associated neurotoxicity syndrome (ICANS), which can be life-threatening and require intensive management [149]. Some patients may also develop a condition known as “hemophagocytic lymphohistiocytosis” (HLH) or “macrophage activation syndrome” (MAS). This is observed in approximately 3.5% of patients treated with CAR-T cells and manifests with fever, hepatosplenomegaly, impaired liver function, and low blood cell counts [150]. Moreover, tumor relapse due to antigen escape, where cancer cells lose or downregulate the targeted antigen, remains a significant hurdle [151]. These challenges highlight the need for continued research to refine CAR-T cell engineering, improve their safety and efficacy, and expand their utility to a wider range of cancers.

The development of ACT targeting NSCLC poses substantial challenges but also significant opportunities for innovation. Key considerations include the identification of optimal tumor-specific antigens (TSAs) or tumor-associated antigens (TAAs) with minimal expression in normal tissues, mitigating off-tumor effects, and overcoming immune tolerance within the TME [152]. Addressing critical issues such as CAR-T cell-associated toxicities, antigen escape, heterogeneity, limited proliferation, and insufficient tumor infiltration requires cutting-edge synthetic biology approaches [153]. Innovations such as multispecific CARs targeting multiple antigens or different cell types (e.g., CD4, CD8, NK cells), spatiotemporally controllable CAR activation using switchable CARs (induced via ultrasound, light, drugs, or adapters), and safety mechanisms, like inducible caspase-9 switches, offer promising solutions [154]. Moreover, self-regulating and self-destructive CAR architectures are being designed to enhance safety, while multi-target CARs aim to address antigen heterogeneity in LC effectively [155]. Another promising avenue involves creating CAR-T cells tailored to specific LC genotypes, leveraging artificial intelligence and CRISPR technology to design synthetic receptors, such as SynNotch systems, that integrate chemical recognition events [156,157]. Researchers are also exploring the mass production of “off-the-shelf” CAR-T cells, potentially simplifying and reducing the costs of CAR-T therapy. The identification of neoantigens and the development of high-throughput screening tools are critical for optimizing CAR-T components and advancing the therapeutic potential of ACT for NSCLC (Table 5).

In summary, clinical and preclinical data indicate that CAR-T cell therapy in NSCLC represents a promising but still experimental platform, where therapeutic benefits are often transient. The future success of this strategy will depend on the development of more sophisticated constructs, combinatorial approaches with ICIs, oncolytic vaccines, or bispecific antibodies, and the identification of predictive biomarkers to better select patients most likely to respond ultimately aiming to enhance the durability and depth of antitumor responses in refractory lung cancer.

### 5.2. Oncological Vaccines

Oncological vaccines represent one of the most promising and innovative frontiers in the fight against cancer, offering new hope, particularly in the treatment of NSCLC. These vaccines are designed to enhance the patient immune system, directing it to selectively recognize and destroy cancer cells, sparing the healthy tissues [158].

Innovative approaches, such as multi-target vaccines that simultaneously target multiple tumor antigens, are showing significant potential. Combinations with immunomodulatory therapies, such as immune checkpoint inhibitors, further enhance treatment efficacy, creating a synergy that could overcome many current limitations [159].

Examples of oncological vaccines already in advanced stages of study include CIMAvax-EGF and TG4010.

The CIMAvax-EGF is a therapeutic vaccine developed in Cuba, designed to block the action of the epidermal growth factor (EGF), a protein that plays a critical role in the growth and proliferation of cancer cells [160]. CIMAvax-EGF aims to stimulate the patient immune system to produce antibodies against EGF, thereby reducing the stimulation of tumor growth. The distinctive feature of this vaccine is its ability to interfere with the binding of EGF to its receptor on the surface of cancer cells, preventing their proliferation [161]. CIMAvax-EGF has been approved in certain countries, such as Cuba, and is used in combination with other treatments, such as chemotherapy, to improve outcomes in patients with advanced NSCLC. Clinical studies have shown that treatment with CIMAvax-EGF can prolong OS, although its efficacy varies depending on the individual characteristics of the tumor and of the patient [162].

The TG4010 is another promising oncological vaccine, developed by the French biotechnology company Transgene [163]. This vaccine is a vector-based on the vaccinia virus, genetically modified to express the MUC1 antigen, a glycoprotein overexpressed on the surface of cancer cells, particularly in NSCLC [164]. TG4010 is designed to stimulate a targeted immune response against cancer cells expressing MUC1. The vaccine is often administered in combination with chemotherapy to enhance the effectiveness of the immune response and reduce tumor size. Clinical trials have demonstrated that TG4010 is well tolerated by patients and that its combination with chemotherapy can improve treatment response and extend survival [165]. Although the results are promising, further studies are required to optimize its clinical application and determine its long-term impact on survival and patients’ quality of life.

Recently, cutting-edge technologies, including the use of genetically modified dendritic cells (DCs) to more effectively activate the immune system, represent a crucial step forward in personalized therapy [166]. A significant milestone in this area is Sipuleucel-T, a DC-based vaccine that received the FDA approval for the treatment of metastatic prostate cancer, in 2010 [167]. Since then, DC vaccines have been the focus of numerous clinical studies. A promising approach involves the intratumoral administration of autologous DC vaccines, such as those modified with the CCL21 gene, to enhance the infiltration of T cells into tumors [168]. This has been demonstrated in studies involving patients with NSCLC. In particular, a Phase I study using AdCCL21-DC showed significant improvements in the activation of CD8+ T cells and the tumor-specific immune response [168,169]. Additionally, combining DC vaccines with immunotherapies, such as pembrolizumab has generated growing interest, as highlighted by ongoing studies.

Moreover, several innovative cancer vaccine strategies are under investigation in NSCLC, aiming to enhance anti-tumor immune responses in various clinical settings. In particular, PDClung01 is a therapeutic vaccine composed of irradiated plasmacytoid DC loaded with multiple HLA-A02:01-restricted tumor-associated peptides (including NY-ESO-1, MAGE-A3/A4, MUC1, survivin, and Melan-A) [170]. In a phase I/II study, patients with stage IV NSCLC (PD-L1 ≥ 50%) treated with high-dose PDClung01 plus pembrolizumab achieved an impressive ORR of 63.2% and a median PFS of 10.9 months, with acceptable safety [170].

Another promising candidate, CAN-2409, is an adenoviral vector carrying HSV-thymidine kinase that, combined with valacyclovir and ICI therapy, acts as an in situ vaccine. In a phase II trial of ICI-resistant NSCLC patients, the combination yielded a median OS of 22 months, with 65% of patients experiencing tumor shrinkage and no dose-limiting toxicities [171].

Additionally, BNT116, an RNA-lipoplex vaccine encoding six tumor antigens, showed an ORR of 35% and a disease control rate of 85% when combined with docetaxel in ICI- and chemotherapy-pretreated NSCLC patients [172]. Similarly, BI1361849, another RNA-based vaccine, was evaluated in combination with durvalumab with or without tremelimumab. While both regimens were well tolerated, the addition of tremelimumab did not improve efficacy, with arm A showing better outcomes (ORR 29%, mPFS 5.7 months) compared to arm B (ORR 11%, mPFS 2.5 months) [173].

Collectively, these studies highlight the potential of cancer vaccines as adjunctive strategies in NSCLC, particularly in overcoming resistance to ICIs. However, despite these promising results, significant challenges remain, including immune tolerance, the limited duration of the immune response, and the complexity of producing DC vaccines. Ongoing research is investigating innovative approaches, including next-generation DC vaccines, the use of different DC types such as plasmacytoid (pDC), conventional (cDC1), and DC-derived exosomes to further improve the efficacy and durability of immune responses against cancer (Table 6) [174,175,176].

### 5.3. Bispecific Antibodies

In recent years, bispecific antibodies (bsAbs) represent a promising innovation in the treatment of NSCLC, particularly in the context of immunotherapy [177]. These antibodies are designed to simultaneously recognize two different molecular targets, enabling a dual modulation of the TME [178]. In the case of NSCLC, the focus is primarily on immune checkpoints such as PD-1/PD-L1 and CTLA-4, which are known to play a crucial role in tumor immune evasion. BsAbs like amivantanab, cadonilimab, and QL1706, which simultaneously target PD-1 and CTLA-4, have shown promising preclinical and clinical results.

For instance, cadonilimab enhances tumor-specific T-cell activation by concurrently blocking these checkpoints while reducing treatment-related toxicity [179]. Clinical studies such as COMPASSION-01 and COMPASSION-03 have explored the efficacy and safety of cadonilimab in patients with advanced solid tumors, including those resistant to chemotherapy and immunotherapy, demonstrating objective responses and acceptable tolerability [180,181]. However, in the specific context of NSCLC, the results have been more variable, with limited objective responses in subgroups of patients with primary or acquired resistance to PD-1/PD-L1 inhibitors.

Another relevant bsAb, QL1706, which combines an anti-PD-1 antibody with an anti-CTLA-4 antibody, has shown anti-tumor activity in advanced solid tumors, including NSCLC, with an improved tolerability profile due to pharmacokinetic optimization and reduced irAEs [182].

Recent clinical data underscore the growing role of amivantamab, a bsAb targeting both EGFR and MET, in the treatment of NSCLC, particularly among patients with EGFR mutations and MET alterations [183]. MET amplification has emerged as a key mechanism of resistance to osimertinib, both in treatment-naïve patients and those with acquired T790M mutations, as well as in cases with de novo MET amplification, supporting the strategy of co-targeting MET early in the treatment course [184]. Amivantamab, developed via the DuoBody platform, is the first and only bsAb approved for NSCLC and has shown significant clinical activity across several patient populations [177]. In the phase 3 PAPILLON trial, the combination of amivantamab and chemotherapy in patients with EGFR exon 20 insertions yielded superior outcomes in response rate (73%) and PFS (11.4 months) compared to chemotherapy alone (6.7 months), leading to its FDA approval as a first-line treatment in this setting [185].

Furthermore, the MARIPOSA phase 3 trial demonstrated that combining lazertinib with amivantamab in treatment-naïve patients with common EGFR mutations significantly improved PFS (23.7 months vs. 16.6 months with osimertinib) [186,187]. This combination has received FDA approval for first-line treatment in patients with EGFR exon 19 deletions or exon 21 L858R mutations. Notably, subgroup analyses revealed consistent PFS benefits among high-risk populations, including those with brain or liver metastases, TP53 co-mutations, and detectable ctDNA. Evidence also suggests intracranial activity, including in patients with leptomeningeal disease [188].

Finally, in patients with uncommon EGFR mutations, the CHRYSALIS-2 study showed encouraging efficacy with lazertinib and amivantamab, both in treatment-naïve and pretreated populations, with ORRs ranging from 48% to 57% and PFS from 7.8 to 19.5 months [189].

Simultaneously, bsAbs combining immunotherapeutic targeting with other pathways are opening new therapeutic avenues.

In this context, ivonescimab (anti-PD-1 and VEGF), combined with chemotherapy, has demonstrated significant anti-tumor activity as a first-line therapy in advanced NSCLC without driver mutations or in patients resistant to prior treatments [190,191]. In particular, in the Phase III HARMONi-2 trial, ivonescimab monotherapy demonstrated a median PFS of 11.1 months, compared to 5.8 months with pembrolizumab, representing a 49% reduction in the risk of disease progression or death (HR = 0.51; *p* < 0.0001) (Table 1). These benefits were consistent across various subgroups, including patients with both low (1–49%) and high (≥50%) PD-L1 expression, as well as those with squamous and non-squamous histology [63]. Regarding toxicity, Ivonescimab displayed a manageable safety profile, with grade ≥ 3 treatment-related adverse events occurring in 29% of patients, the most frequent being hypertension (5%). Serious treatment-related adverse events were reported in 21% of cases [63]. Based on these encouraging outcomes, ivonescimab has been approved in China in combination with chemotherapy for the treatment of EGFR-TKI-resistant SqCC and non-SqCC NSCLC, and is currently under priority review as a first-line monotherapy for PD-L1-positive NSCLC [192]. These findings underscore the potential of bsAbs not only to enhance immune responses but also to overcome resistance limiting the effectiveness of traditional monotherapies.

A new frontier, still under investigation, could be represented by new generations of bsAbs, including the Bispecific T-Cell Engagers (BiTEs) and the Bispecific Nanobodies (BsNb). These two types of bsAbs have been emerging as promising strategies for the treatment of NSCLC. In particular, BiTEs are designed to recruit and activate cytotoxic T cells by simultaneously binding to a tumor antigen, such as EGFR or MET, and the CD3 receptor on T cells, facilitating targeted activation and tumor cell lysis [193]. On the other hand, Bispecific Nanobodies, derived from single-domain antibodies, offer additional advantages due to their smaller size, which allows for better tumor penetration and faster systemic clearance, improving their safety profile [194]. These new approaches are particularly promising in the immunosuppressive microenvironment of NSCLC, where they can overcome resistance to ICIs and amplify the antitumor response.

Despite these advances, significant challenges remain, such as identifying predictive biomarkers of response and managing associated toxicity. Ongoing clinical trials, such as AK104, SSGJ-707, and MCLA-128 are helping to better define the role of these agents in NSCLC therapy and establish new therapeutic paradigms (Table 7). If confirmed, bsAbs could represent a milestone in NSCLC immunotherapy, offering more effective and personalized options for patients.

## 6. Conclusions

The development of advanced diagnostic tests with greater specificity in identifying and characterizing oncological diseases has facilitated a shift from a traditional, generalized clinical model to a personalized diagnostic and therapeutic approach. This evolution enables tailored patient care, providing increasingly accurate and effective therapeutic strategies. In this context, molecular profiling plays a fundamental role, offering crucial information not only at diagnosis but also throughout disease evolution, thus supporting dynamic and evidence-based treatment decisions. This approach represents a model to follow, because it will provide the patient with greater possibilities of receiving increasingly effective therapies, minimizing adverse events due to the administration of non-targeted therapies, with consequent undoubted clinical benefits for the patient.

The past ten years have witnessed the development of the immunotherapy based on specific mAbs, the ICIs. When ICIs target the immune-checkpoint of the T cell reaction, the host immune system is reactivated and better able to identify and eliminate NSCLC cells. The ICIs have been demonstrated to be successful in lowering tumor burden and enhancing the prognosis of LC patients, either alone or in combination with other ICIs, conventional chemotherapy. However, one of the main disadvantages is the resistance that some patients develop to these therapies, reducing long-term efficacy. Recently, there has been growing interest in clinical trials, particularly focusing on innovative immunotherapeutic approaches. These include the use of new immune checkpoint inhibitors targeting different pathways to avoid resistance, and CAR-T cell therapy, which could potentially offer more targeted and personalized responses. Additionally, oncological vaccines and bispecific antibodies are the subject of intense studies, as they may stimulate the immune system to recognize specific tumor antigens, providing new therapeutic options for patients who do not respond to conventional therapies. However, although preliminary data from early-phase trials have shown encouraging signals of efficacy, several late-phase studies have yielded disappointing or inconclusive results, tempering initial optimism. These inconsistencies highlight the complexity of tumor–immune system interactions and underscore the need for more refined patient selection and biomarker-driven approaches. Finally, while immunotherapy continues to represent a cornerstone of NSCLC treatment, its future success will depend on overcoming resistance mechanisms and validating these novel approaches through robust clinical evidence.

## Figures and Tables

**Figure 1 ijms-26-11055-f001:**
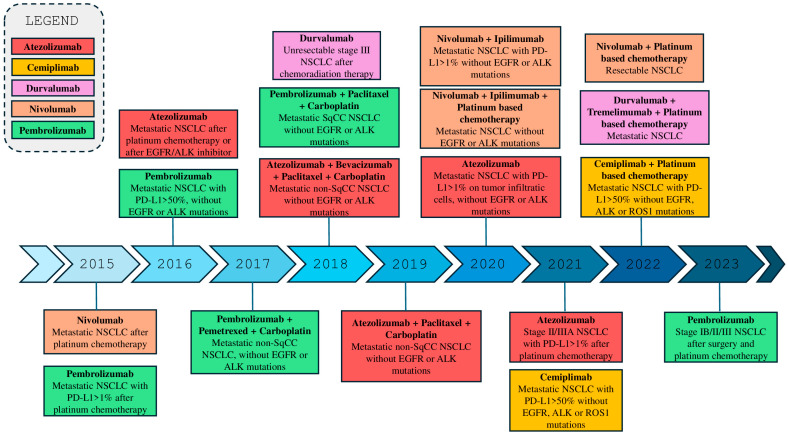
Timeline of the FDA approval of Immune Checkpoint Inhibitors (ICIs) for NSCLC treatment.

**Figure 2 ijms-26-11055-f002:**
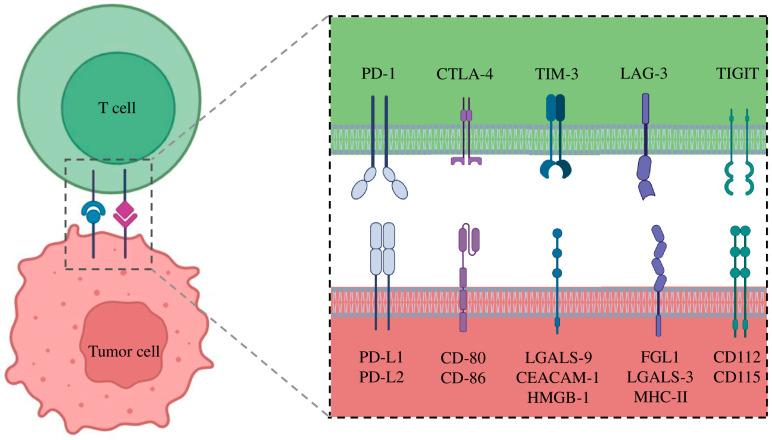
Interaction of Immune Checkpoints and their corresponding ligands.

**Figure 3 ijms-26-11055-f003:**
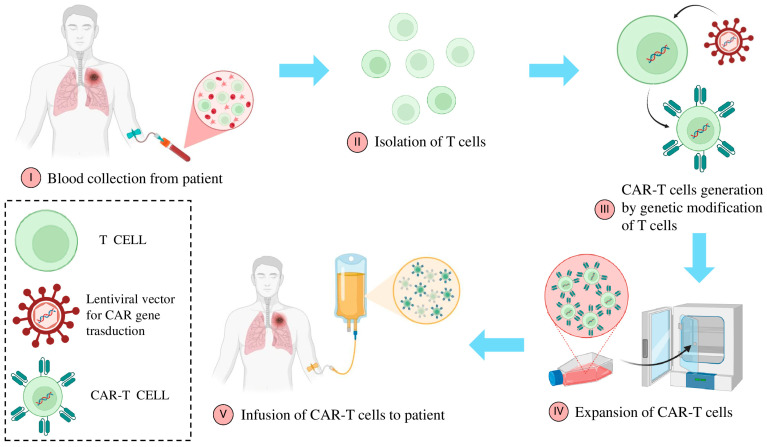
Autologous CAR-T cell manufacturing process. The production of autologous CAR T cells starts with the patient’s leukapheresis, followed by the enrichment and activation of T cells. The activated T lymphocytes are then transduced, for instance, using a lentiviral vector, to enable the introduction and, in some cases, permanent integration of the CAR transgene. The genetically modified T cells are subsequently expanded in either static or dynamic systems to reach the required therapeutic dose, cryopreserved, and reinfused into the patient.

**Table 2 ijms-26-11055-t002:** Ongoing clinical studies on anti-Lag3 therapies for NSCLC registered at clinicaltrials.gov (accessed on 10 November 2025).

NCT Number	Interventions	Study Status	Sponsor	Phase
NCT03916627	Fianlimab	Active not recruiting	Regeneron Pharmaceuticals	Phase 2
NCT05785767	Fianlimab	Recruiting	Regeneron Pharmaceuticals	Phase 2|3
NCT05800015	Fianlimab	Recruiting	Regeneron Pharmaceuticals	Phase 2|3
NCT06161441	Fianlimab	Recruiting	Regeneron Pharmaceuticals	Phase 2
NCT06865339	Fianlimab	Recruiting	Nitin Ohri	Phase 2
NCT06918132	Fianlimab	Recruiting	Mayo Clinic	Phase 2
NCT05978401	GLS-012	Not yet recruiting	Guangzhou Gloria Biosciences	Phase 1|2
NCT05787613	HLX26	Active not recruiting	Shanghai Henlius Biotech	Phase 2
NCT04205552	Relatlimab	Recruiting	University Hospital, Essen	Phase 2
NCT04623775	Relatlimab	Active not recruiting	Bristol-Myers Squibb	Phase 2
NCT05176483	Relatlimab	Active not recruiting	Exelixis	Phase 1
NCT06561386	Relatlimab	Recruiting	Bristol-Myers Squibb	Phase 3

**Table 3 ijms-26-11055-t003:** Ongoing clinical studies on anti-Tim3 therapies for NSCLC registered at clinicaltrials.gov (accessed on 10 November 2025).

NCT Number	Interventions	Study Status	Sponsor	Phase
NCT02817633	TSR-022	Active not recruiting	Tesaro, Inc.	Phase 1
NCT04931654	AZD7789	Active not recruiting	AstraZeneca	Phase 1|2
NCT06162572	S095018	Recruiting	Servier Bio-Innovation LLC	Phase 1|2

**Table 4 ijms-26-11055-t004:** Ongoing clinical studies on anti-TIGIT therapies for NSCLC registered at clinicaltrials.gov (accessed on 10 November 2025).

NCT Number	Study Status	Interventions	Sponsor	Phase
NCT03563716	Active not recruiting	Tiragolumab	Genentech, Inc.	Phase 2
NCT04294810	Active not recruiting	Tiragolumab	Hoffmann-La Roche	Phase 3
NCT04374877	Active not recruiting	CHS-388	Coherus Biosciences, Inc.	Phase 1
NCT04736173	Active not recruiting	Domvanalimab	Arcus Biosciences, Inc.	Phase 2
NCT04746924	Active not recruiting	Ociperlimab	BeiGene	Phase 3
NCT04995523	Recruiting	AZD2936	AstraZeneca	Phase 1|2
NCT05102214	Recruiting	HLX301	Shanghai Henlius Biotech	Phase 1|2
NCT05417321	Recruiting	HB0036	Shanghai Huaota Biopharmaceutical	Phase 1|2
NCT05676931	Active not recruiting	Domvanalimab	Gilead Sciences	Phase 2
NCT06627647	Recruiting	Rilvegostomig	AstraZeneca	Phase 3
NCT06692738	Recruiting	Rilvegostomig	AstraZeneca	Phase 3
NCT06773507	Recruiting	BC008-1A	Sichuan Luzhou Buchang Biopharmaceutical	Phase 1

**Table 5 ijms-26-11055-t005:** Ongoing clinical studies on CAR-T cell therapies for NSCLC registered at clinicaltrials.gov (accessed on 10 November 2025).

NCT Number	Study Status	Interventions	Sponsor	Phase
NCT04503278	Recruiting	CLDN6 CAR-T	BioNTech Cell & Gene Therapies GmbH	Phase 1
NCT04556669	Recruiting	aPD-L1 and anti-CD22 CAR-T	Hebei Senlang Biotechnology Inc., Ltd.	Phase 1
NCT05060796	Recruiting	CXCR5 modified EGFR CART	Hospital of Guangzhou Medical University	Phase 1
NCT05239143	Active not recruiting	P-MUC1C-ALLO1 CAR-T	Poseida Therapeutics, Inc.	Phase 1
NCT05620342	Recruiting	iC9.GD2.CAR-T	UNC Lineberger Comprehensive Cancer Center	Phase 1
NCT05736731	Active not recruiting	A2B530	A2 Biotherapeutics Inc.	Phase 1|2
NCT06043466	Recruiting	CEA-targeted CAR-T	Chongqing Precision Biotech Co., Ltd.	Phase 1
NCT06051695	Recruiting	A2B694	A2 Biotherapeutics Inc.	Phase 1|2
NCT06653023	Recruiting	Universal CAR-T injection	Wondercel Biotech (ShenZhen)	Phase 1
NCT06682793	Recruiting	A2B395	A2 Biotherapeutics Inc.	Phase 1|2
NCT06972576	Recruiting	EphA2-targeted CAR-T Cells	Zhejiang University	Phase 1
NCT07116057	Recruiting	MOv19-BBz CAR T cells	University of Pennsylvania	Phase 1

**Table 6 ijms-26-11055-t006:** Ongoing clinical studies on oncological vaccines for NSCLC registered at clinicaltrials.gov (accessed on 10 November 2025).

NCT Number	Study Status	Interventions	Sponsor	Phase
NCT01720836	Recruiting	MUC1 Peptide Vaccine	Olivera Finn	Phase 1|2
NCT02432963	Active not recruiting	Virus Ankara Vaccine Expressing p53	City of Hope Medical Center	Phase 1
NCT03546361	Active not recruiting	Autologous DC-Adenovirus CCL21 Vaccine	Jonsson Comprehensive Cancer Center	Phase 1
NCT03552718	Active not recruiting	YE-NEO-001	NantBioScience, Inc.	Phase 1
NCT03908671	Recruiting	Personalized mRNA Tumor Vaccine	Stemirna Therapeutics	Not Applicable
NCT03970746	Active not recruiting	DC vaccine	PDC*line Pharma SAS	Phase 1|2
NCT04147078	Recruiting	DC vaccine	Sichuan University	Phase 1
NCT04266730	Not yet recruiting	PANDA-VAC	UNC Lineberger Comprehensive Cancer Center	Phase 1
NCT04298606	Recruiting	EGF-rP64K/Montanide ISA 51 Vaccine	Roswell Park Cancer Institute	Phase 1
NCT04503278	Recruiting	CLDN6 uRNA-LPX/CLDN6 modRNA-LPX	BioNTech Cell & Gene Therapies GmbH	Phase 1
NCT04686305	Recruiting	T-DXd	AstraZeneca	Phase 1
NCT05104515	Recruiting	OVM-200	Oxford Vacmedix UK Ltd.	Phase 1
NCT05142189	Recruiting	BNT116	BioNTech SE	Phase 1
NCT05195619	Active not recruiting	PEP-DC vaccine	Centre Hospitalier Universitaire Vaudois	Phase 1
NCT05242965	Active not recruiting	Plasmid DNA Vaccine	University of Washington	Phase 2
NCT05254184	Recruiting	Mutant KRAS-Targeted Peptide Vaccine	Sidney Kimmel Comprehensive Cancer Center	Phase 1
NCT05269381	Recruiting	Neoantigen Peptide Vaccine	Mayo Clinic	Phase 1|2
NCT05344209	Recruiting	UV1	Vestre Viken Hospital Trust	Phase 2
NCT05557591	Active not recruiting	BNT116	Regeneron Pharmaceuticals	Phase 2
NCT05950139	Recruiting	Peptide vaccine	Sidney Kimmel Comprehensive Cancer Center	Phase 1|2
NCT06015724	Recruiting	KRAS vaccine	Georgetown University	Phase 2
NCT06095934	Recruiting	Neoantigen vaccine	Nanjing Drum Tower Hospital of Nanjing University	Not Applicable
NCT06253520	Recruiting	GRT-C903/GRT-R904	National Cancer Institute (NCI)	Phase 1
NCT06472245	Recruiting	OSE2101	OSE Immunotherapeutics	Phase 3
NCT06685653	Not yet recruiting	RGL-270	Nanjing Tianyinshan Hospital	Phase 1
NCT06735508	Not yet recruiting	mRNA Neoantigen Vaccine	Guangdong Provincial People’s Hospital	Phase 1
NCT06751849	Recruiting	Neoantigen-loaded DC vaccine	Hospital of Nanchang University	Phase 2
NCT06751901	Recruiting	Neoantigen-based peptide vaccine	Hospital of Nanchang University	Phase 2
NCT06752044	Recruiting	Neoantigen-based peptide vaccine	Hospital of Nanchang University	Not Applicable
NCT06752057	Recruiting	Neoantigen-loaded DC vaccine	Hospital of Nanchang University	Not Applicable
NCT07073183	Not yet recruiting	CV09070101 mRNA vaccine	CureVac	Phase 1

**Table 7 ijms-26-11055-t007:** Ongoing clinical studies on bispecific antibody therapies for NSCLC registered at clinicaltrials.gov (accessed on 10 November 2025).

NCT Number	Study Status	Interventions	Sponsor	Phase
NCT02609776	Active not recruiting	Amivantamab	Janssen Research & Development, LLC	Phase 1
NCT02912949	Active not recruiting	MCLA-128	Merus N.V.	Phase 2
NCT03526835	Recruiting	MCLA-158	Merus N.V.	Phase 1|2
NCT03797391	Recruiting	EMB-01	Shanghai EpimAb Biotherapeutics	Phase 1|2
NCT04140500	Active not recruiting	RO7247669	Hoffmann-La Roche	Phase 1|2
NCT04603287	Active not recruiting	SI-B001	Sichuan Baili Pharmaceutical	Phase 1
NCT04606472	Active not recruiting	SI-B003	Sichuan Baili Pharmaceutical	Phase 1
NCT04777084	Recruiting	IBI318	Hunan Province Tumor Hospital	Phase 1
NCT04868877	Recruiting	MCLA-129	Merus N.V.	Phase 1|2
NCT04930432	Recruiting	MCLA-129	Betta Pharmaceuticals	Phase 1|2
NCT04931654	Active not recruiting	AZD7789	AstraZeneca	Phase 1|2
NCT04995523	Recruiting	AZD2936	AstraZeneca	Phase 1|2
NCT05102214	Recruiting	HLX301	Shanghai Henlius Biotech	Phase 1|2
NCT05117242	Active not recruiting	Acasunlimab	Genmab	Phase 2
NCT05180474	Active not recruiting	GEN1047	Genmab	Phase 1|2
NCT05360381	Active not recruiting	HLX35	Shanghai Henlius Biotech	Phase 1
NCT05377658	Recruiting	AK104	Henan Cancer Hospital	Phase 2
NCT05420220	Recruiting	KN046	Jiangsu Alphamab Biopharmaceuticals	Phase 2
NCT05498389	Not yet recruiting	EMB-01	Shanghai EpimAb Biotherapeutics	Phase 1|2
NCT05663866	Active not recruiting	Amivantamab	Janssen Research & Development, LLC	Phase 2
NCT05780307	Recruiting	IMM2520	ImmuneOnco Biopharmaceuticals	Phase 1
NCT05816499	Active not recruiting	Cadonilimab	Shanghai Chest Hospital	Phase 1|2
NCT05845671	Recruiting	Amivantamab	University of Colorado, Denver	Phase 1|2
NCT06015568	Not yet recruiting	MCLA-129	Betta Pharmaceuticals	Phase 1
NCT06116682	Recruiting	Amivantamab	SWOG Cancer Research Network	Phase 2
NCT06147037	Recruiting	FPI-2053	Fusion Pharmaceuticals Inc.	Phase 1
NCT06196814	Not yet recruiting	AK112	Hunan Province Tumor Hospital	Phase 1|2
NCT06361927	Recruiting	SSGJ-707	Sunshine Guojian Pharmaceutical	Phase 2
NCT06412471	Recruiting	SSGJ-707	Sunshine Guojian Pharmaceutical	Phase 2
NCT06417008	Recruiting	HS-20117	Hansoh BioMedical R&D Company	Phase 2|3
NCT06424821	Recruiting	Cadonilimab	Shanghai Pulmonary Hospital	Phase 2
NCT06467500	Recruiting	Cadonilimab	Xin-Hua Xu	Phase 2
NCT06532591	Recruiting	Cadonilimab	Sichuan Cancer Hospital and Research Institute	Phase 2
NCT06621563	Recruiting	HS-20117	Hansoh BioMedical R&D Company	Phase 1
NCT06724263	Not yet recruiting	B1962	Tasly Biopharmaceuticals	Phase 2
NCT06766591	Not yet recruiting	Ivonescimab	Jiangsu Province Nanjing Brain Hospital	Not Applicable
NCT06793813	Recruiting	Cadonilimab	Chinese Academy of Medical Sciences	Phase 2
NCT06943820	Recruiting	AK129	Akeso	Phase 1|2
NCT06996782	Recruiting	Rilvegostomig	AstraZeneca	Phase 1|2

## Data Availability

The original contributions presented in this study are included in the article. Further inquiries can be directed at the corresponding authors.

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
