# Peer review of "Directions of Immunotherapy for Non-Small-Cell Lung Cancer Treatment: Past, Present and Possible Future"

_ijms, 2025, doi:10.3390/ijms262211055_

Round 1

Reviewer 1 Report

Comments and Suggestions for Authors

This is a very comprehensive, well-structured, and timely review of the immunotherapy landscape for Non-Small Cell Lung Cancer (NSCLC). The authors have successfully navigated a vast and rapidly evolving field, presenting the information in a logical "past, present, and future" framework that is both accessible to newcomers and a valuable summary for experts. The review adeptly covers the foundational immune checkpoint inhibitors (ICIs), the current standard-of-care combination therapies, and a wide array of emerging modalities including novel checkpoint targets, cellular therapies, vaccines, and bispecific antibodies. The extensive use of summary tables for ongoing clinical trials is up to date. Overall, this is a high-quality manuscript that will serve as an excellent reference for clinicians and researchers in oncology and pharmaceutical sciences. I would recommend acceptance with minor revisions. 

  1. The section on TIGIT inhibitors is slightly too optimistic and could benefit from a more balanced critical perspective. Some high-profile Phase III trial failures (e.g., SKYSCRAPER studies) have tempered enthusiasm for this target class across the field. Adding a sentence to acknowledge that this early promise has not yet been realized in pivotal NSCLC trials would provide a more accurate and balanced picture.
  2. For Section 5, try using a more encompassing title other than "Noval Immune Checkpoint Targets" as cancer vaccines and CAR-T therapies target not just immune checkpoints.

Author Response

We appreciate the reviewers’ time and suggestions for improving the paper. The changes in the manuscript are marked by the “Track Changes” function. The specific responses to the reviewer’s comments are listed below.

  1. The section on TIGIT inhibitors is slightly too optimistic and could benefit from a more balanced critical perspective. Some high-profile Phase III trial failures (e.g., SKYSCRAPER studies) have tempered enthusiasm for this target class across the field. Adding a sentence to acknowledge that this early promise has not yet been realized in pivotal NSCLC trials would provide a more accurate and balanced picture.

Response: Thank you for raising this important point. We have added it in chapter “4.3. T cell immunoreceptor with immunoglobulin and ITIM domain” of the revised manuscript (Lines 599-613 in the revised manuscript):

“Despite the initial promise of TIGIT blockade, recent Phase II/III results have been less encouraging. Notably, The Phase III SKYSCRAPER-01 trial evaluated the combination of the anti-TIGIT monoclonal antibody tiragolumab with the anti–PD-L1 mAb atezolizumab as first-line therapy for patients with PD-L1–high (≥50%) advanced or metastatic NSCLC [104]. Although the combination showed a numerical improvement in both PFS (median 7.0 vs. 5.6 months; HR = 0.78) and OS (median 23.1 vs. 16.9 months; HR = 0.87) compared with atezolizumab monotherapy, these differences did not reach statistical significance [104]. Similarly, subsequent analyses did not confirm the early efficacy signals observed in CITYSCAPE study. Indeed, in 2024 the Phase II/III SKYSCRAPER-06 trial failed to demonstrate a statistically significant improvement in OS or PFS with tiragolumab plus atezolizumab and chemotherapy compared to pembrolizumab in combination with chemotherapy in non-SqCC NSCLC patients [105]. These outcomes have tempered the initial enthusiasm for TIGIT-targeted immunotherapy, underscoring the complexity of checkpoint interactions and the need for refined biomarker-driven strategies to identify responders.”

  1. For Section 5, try using a more encompassing title other than "Noval Immune Checkpoint Targets" as cancer vaccines and CAR-T therapies target not just immune checkpoints.

Response: As suggested, we have modified the title of Section 5 as follow “Novel approaches to NSCLC immunotherapy”

Reviewer 2 Report

Comments and Suggestions for Authors

In this manuscript Leone GM et al. provide a comprehensive and excellent review on the history and current status of immunotherapy for non-small cell lung cancer.

The manuscript is well organized and pleasant to read, supported by very well-made figures and extremely informative tables, which summarize the clinical trials relevant to this topic. The Readers will greatly benefit from this article, since it will immediately bring them up-to-speed, which should be indeed the goal of any review papers.

Here are some minor comments that will further improve an already outstanding manuscript:

  1. Line 40. Please add the current OS for each NSCLC stage.
  2. Paragraph 2 has only one sub-paragraph (2.1 - PD1/PD-L1 axis). To be consistent with what anticipated in the introduction to this paragraph, I suggest adding another sub-paragraph (2.2) to briefly talk about the CTLA-4 pathway.
  3. Line 111. Please take out the parentheses from KEYNOTE-024 (it reads much more logically).
  4. All tables need to be updated from May 10, 2025, as needed.
  5. Line 321. For consistency, please indicate that cobolimab is an IgG4 antibody.
  6. Line 337. Please correct the percentage number.
  7. Line 342. Sabatolimab should not be written with capital "S".
  8. Line 365. Please provide a reference to the sentence: "In NSCLC, overexpression.... reduced cytotoxicity".
  9. Lines 410 and 414, please indicate that magrolimab and monalizumab are IgG4 antibodies.

Author Response

We appreciate the reviewers’ time and suggestions for improving the paper. The changes in the manuscript are marked by the “Track Changes” function. The specific responses to the reviewer’s comments are listed below.

  1. Line 40. Please add the current OS for each NSCLC stage.

Response: We have revised the manuscript as suggested (Lines 39-46 in the revised manuscript):

“LC outcomes are strongly influenced by the stage at diagnosis. Patients diagnosed at an early stage generally have a favorable prognosis, with 5-year survival rates of approximately 70–90% for Stage I and 50–70% for Stage II [14]. However, survival decreases markedly in more advanced stages, reaching 30–35% for Stage III and only about 9–10% for Stage IV. Unfortunately, LC is most often diagnosed at an advanced stage, when curative options are limited and systemic drug therapy, guided by the immunological and molecular profile of the tumor, represents the standard of care [15–17]”

  1. Paragraph 2 has only one sub-paragraph (2.1 - PD1/PD-L1 axis). To be consistent with what anticipated in the introduction to this paragraph, I suggest adding another sub-paragraph (2.2) to briefly talk about the CTLA-4 pathway.

Response: Thank you for pointing this out. We have now discussed about CTLA-4 pathway in chapter 2.2. (Lines 234-271 in the revised manuscript):

  1. Line 111. Please take out the parentheses from KEYNOTE-024 (it reads much more logically).

Response: We have removed the parentheses from KEYNOTE-024  as suggested (Line 157 in the revised manuscript)

  1. All tables need to be updated from May 10, 2025, as needed.

Response: We have updated all tables as of November 10

  1. Line 321. For consistency, please indicate that cobolimab is an IgG4 antibody.

Response: We have indicated it on line 519 of the revised manuscript

  1. Line 337. Please correct the percentage number.

Response: We have corrected it on line 536 of the revised manuscript

  1. Line 342. Sabatolimab should not be written with capital "S".

Response: We have corrected it on line 541 of the revised manuscript

  1. Line 365. Please provide a reference to the sentence: "In NSCLC, overexpression.... reduced cytotoxicity".

Response: We have added the reference to the sentence on line 572 of the revised manuscript

  1. Lines 410 and 414, please indicate that magrolimab and monalizumab are IgG4 antibodies

Response: We have specified it on lines 642 and 646 of the revised manuscript

Reviewer 3 Report

Comments and Suggestions for Authors

The present manuscript present its strength through a timely and important topic with a comprehensive scope. It gave a broad and recent literature coverage and appropriate emphasis on personalized/precision approaches. However, some weakness should be addressed first as following:

  1. The review seemed much like a broad summary rather than providing novel synthesis or clear actionable conclusions for researchers. Please clarify the manuscript’s unique contribution in the Introduction.
  2. There was no methods section describing how literature was identified, selected, and synthesized, including databases searched, date range, inclusion/exclusion criteria, search terms, etc.
  3. The present manuscript showed repetition and rough transitions, for example“6. Conclusions 6. Conclusions”, so I suggest the authors to reorganize into clear sections.
  4. Although the manuscript listed trials and agents, it was lack of critical appraisal. For key pivotal trials, like CheckMate series, KEYNOTE series, IMpower, HARMONi‑2, etc., please include a concise critical appraisal, including trial design, population, endpoints, magnitude of benefit, safety signals, and limitations.
  5. The biomarker section mentioned PD‑L1, TMB, TILs but didn’t critically address limitations, assay variability, and cutoff inconsistency. Please discuss more about PD‑L1 assay heterogeneity and impact on clinical decision making.
  6. Based on present review, resistance was acknowledged but discussion was too  Please expand on known mechanisms like primary vs acquired resistance and tie these to therapeutic strategies and cite more recent mechanistic and translational studies to identify gaps for future research.
  7. Even though emerging modalities have been described optimistically, it still needs a balanced view of realistic challenges, such as antigen heterogeneity and trafficking and persistence in solid tumors. For CAR‑T and bispecifics, I think it’s better to include current early‑phase efficacy data, major challenges, and specific engineered solutions.
  8. The authors listed some references about irAEs, but didn’t show sufficient discussion on incidence, management, and impact on outcomes about
  9. The conclusions seemed too optimistic without enough evidence support.

Author Response

We appreciate the reviewers’ time and suggestions for improving the paper. The changes in the manuscript are marked by the “Track Changes” function. The specific responses to the reviewer’s comments are listed below.

  1. The review seemed much like a broad summary rather than providing novel synthesis or clear actionable conclusions for researchers. Please clarify the manuscript’s unique contribution in the Introduction.

Response: We have modified it in the Introduction of the revised manuscript

  1. There was no methods section describing how literature was identified, selected, and synthesized, including databases searched, date range, inclusion/exclusion criteria, search terms, etc.

Response: The comprehensive literature search was conducted using the PubMed, Scopus, and Web of Science databases to identify relevant English-language publications available up to 2025. The search terms included “non-small cell lung cancer”, “immunotherapy”, “immune checkpoint inhibitors”, “adoptive T-cell therapy”, “cancer vaccines”, and “bispecific antibodies”. Additional references were identified through citation tracking of key articles. Studies were selected based on their relevance to the topic, novelty, and clinical significance.

  1. The present manuscript showed repetition and rough transitions, for example“6. Conclusions 6. Conclusions”, so I suggest the authors to reorganize into clear sections.

Response: As suggested, we have removed repetitions and improved the connections between paragraphs

  1. Although the manuscript listed trials and agents, it was lack of critical appraisal. For key pivotal trials, like CheckMate series, KEYNOTE series, IMpower, HARMONi‑2, etc., please include a concise critical appraisal, including trial design, population, endpoints, magnitude of benefit, safety signals, and limitations.

Response: We have added Table 1, which includes the main information of the key trials. Additionally, We have also critically analysed the results of the trials

  1. The biomarker section mentioned PD‑L1, TMB, TILs but didn’t critically address limitations, assay variability, and cutoff inconsistency. Please discuss more about PD‑L1 assay heterogeneity and impact on clinical decision making.

Response: We have added it in lines 69-102 of the revised manuscript

“Recent studies are focusing on identifying predictive biomarkers, such as PD-L1 expression, tumor mutational burden (TMB) and Tumor-infiltrating lymphocytes (TILs), and on optimizing therapeutic combinations to further improve the outcomes and to reduce adverse drug reactions [33–35]. However, despite their central role in guiding immunotherapy, these biomarkers showed significant limitations. In particular, PD-L1 expression is highly heterogeneous both spatially and temporally, varying between primary and metastatic sites and even within the same tumor region [36]. Its assessment is further complicated by inter-assay variability among immunohistochemical tests (e.g., 22C3, 28-8, SP263, SP142) and the lack of standardized cutoff values, which were historically defined in trial-specific contexts rather than through biologically consistent thresholds [37]. Similarly, TMB, though associated with improved response to immune checkpoint inhibitors, lacks a universally accepted assay or cutoff, and its predictive value may differ depending on sequencing platform, bioinformatic pipeline, and tumor subtype [38,39]. Moreover, TMB does not fully capture the immunogenic landscape, since not all mutations generate neoantigens capable of eliciting an immune response [40]. The use of TILs as a biomarker also faces methodological challenges due to subjective interpretation, intratumoral variability, and absence of validated quantification standards in lung cancer [41]n. Collectively, these factors limit the reliability and reproducibility of current biomarkers and underscore the need for integrated, multi-parametric approaches combining molecular, histopathological, and immunological data to better predict immunotherapy outcomes in NSCLC.

This review explores the rationale and current evidence supporting approved immunotherapy strategies for advanced and metastatic NSCLC, integrating insights from preclinical research and ongoing clinical trials. Particular attention is given to emerging combination strategies aimed at overcoming resistance and improving clinical outcomes. Furthermore, novel therapeutic approaches, current challenges, and future perspectives in the evolving landscape of NSCLC immunotherapy will be discussed.”

  1. Based on present review, resistance was acknowledged but discussion was too  Please expand on known mechanisms like primary vs acquired resistance and tie these to therapeutic strategies and cite more recent mechanistic and translational studies to identify gaps for future research.

Response: We have discussed it in chapter 5 of the revised manuscript

  1. Even though emerging modalities have been described optimistically, it still needs a balanced view of realistic challenges, such as antigen heterogeneity and trafficking and persistence in solid tumors. For CAR‑T and bispecifics, I think it’s better to include current early‑phase efficacy data, major challenges, and specific engineered solutions.

Response: We added it in chapter 5.1 of the revised manuscript

  1. The authors listed some references about irAEs, but didn’t show sufficient discussion on incidence, management, and impact on outcomes about

Response: We have revised it in lines 668-686 of the revised manuscript

  1. The conclusions seemed too optimistic without enough evidence support.

Response: As suggested, we modified the conclusion